# BRAF Controls the Effects of Metformin on Neuroblast Cell Divisions in *C. elegans*

**DOI:** 10.3390/ijms22010178

**Published:** 2020-12-26

**Authors:** Zhi Qu, Shaoping Ji, Shanqing Zheng

**Affiliations:** 1School of Nursing and Health, Henan University, Kaifeng 475004, China; 10210056@henu.edu.cn; 2Medical School, Henan University, Kaifeng 475004, China; shaopingji@henu.edu.cn; 3School of Basic Medical Sciences, Henan University, Kaifeng 475004, China

**Keywords:** *C. elegans*, metformin, neuronal cell divisions, L1 arrest, BRAF, MPK-1

## Abstract

Metformin has demonstrated substantial potential for use in cancer treatments. Liver kinase B (LKB)-AMP-activated protein kinase (AMPK) and mTOR are reported to be the main targets of metformin in relation to its ability to prevent cancer cell proliferation. However, the role of metformin in the control of neoplastic cancer cell growth is possibly independent of LKB-AMPK and mTOR. Using *C. elegans* as a model, we found that the neuronal Q-cell divisions in L1-arrested worms were suppressed following metformin treatment in AMPK-deficient mutants, suggesting that the mechanism by which metformin suppresses these cell divisions is independent of AMPK. Our results showed that the mTOR pathway indeed played a role in controlling germ cell proliferation, but it was not involved in the neuronal Q-cell divisions occurring in L1-arrested worms. We found that the neuronal Q-cells divisions were held at G1/S cell stage by metformin *in vivo*. Additionally, we demonstrated that metformin could reduce the phosphorylation activity of BRAF and block the BRAF-MAPK oncogenesis pathway to regulate neuronal Q-cell divisions during L1 arrest. This work discloses a new mechanism by which metformin treatment acts to promote neuronal cancer prevention, and these results will help promote the study of the anticancer mechanisms underlying metformin treatments.

## 1. Introduction

Metformin, a first-line drug used in the treatment of type 2 diabetes, is also reported to have potential benefits in cancer prevention and treatment [1,2,3,4]. Metformin likely inhibits the mitochondrial electron transport chain Complex I, resulting in changes to the AMP/ATP ratio, which can activate the AMP/ATP energy-sensor AMP-activated protein kinase (AMPK) signal pathway [5,6]. Alternatively, the AMPK pathway might also be activated from upstream by metformin via the liver kinase B1 (LKB1) protein [7,8]. However, there are also a number of studies suggesting that the function of metformin is independent of the AMPK pathway [9,10,11,12,13,14,15,16,17]. The functions of metformin in relation to age-related diseases, metabolic syndrome, cognitive disorders, and canner are also likely connected to the mTOR pathway [3,4,5,6,7,8,9,10,11,12,13,14,15,16,17,18]. Evidence suggests that inhibition of the mTOR pathway by metformin is both dependent on and independent of AMPK activation [18]. AMPK phosphorylates tuberous sclerosis complex protein 2 (TSC2), which inhibits mTORC1, leading to a decrease in protein synthesis and cell growth [19]. Inhibition of mTOR by metformin independent of AMPK activation has also been demonstrated in mice lung cancer cells [20]. Metformin can also activate mTORC1 via lysosome-localized Rag GTPases directly bound to the Raptor subunit of mTORC1 [3,18]. These studies suggest that the function of metformin might govern lysosome-LKB-AMPK and Rag-mTORC1 to control cancer cell proliferation. However, the antineoplastic cancer potential of metformin has also been assessed and it appears that this activity is unrelated to AMPK-dependent and -independent inhibition of mTOR [12,21,22,23]. As such, metformin might employ mechanisms other than the AMPK and mTOR pathways.

*C. elegans* is widely used as a model to study oncogenic gene regulation and drug function. During hermaphrodite development, if the newly hatched L1 worms have no available food, the worms will undergo a quiescence state called an L1 arrest [24]. During L1 arrest, worms do not undertake any postembryonic development, including cell divisions and growth. As such, L1-arrested worms provide an important model to study the functions of drugs and chemicals on the genetic circuits regulating cell proliferation and development. Previously, ours and other groups have shown that loss of AMPK and PTEN in *C. elegans* causes L1-arrested worms to undergo neuronal Q-cell and germ-cell proliferation [25,26]. AMPK deficiency results in such cell proliferations even in the absence of food, which provides an ideal model to study cell development and drug interventions.

The results of this current study demonstrated that metformin could suppress the aberrant Q-cell divisions occurring in L1-arrested worms deficient in AMPK. As mentioned above, AMPK loss-of-function can activate the mTOR pathway, but it was observed that metformin did not act through mTOR to suppress these neuronal cell divisions, even though it has been shown that the mTOR pathway is involved in regulating germ-cell proliferation during L1 arrest [25]. Previous work has shown that AMPK can act through RAF-MAPK to control neuronal Q-cell divisions during L1 arrest [26]. RAF-MAPK is a pivotal oncogenic pathway conserved in animals and is also regulated by AMPK in *C. elegans*, mice, and humans to control cell proliferation and cancer development [26,27,28]. AMPK can phosphorylate the RAF family protein BRAF, resulting in attenuation of the MEK-ERK signal in mammals [29]. BRAF can also be regulated by another ontogenetic protein, PP2A, to control the MEK-ERK pathway [30,31]. This current study found that metformin could suppress cell proliferation even when AMPK was abolished, while the loss of BRAF eliminated the suppressive function of metformin towards the Q-cell divisions. As such, metformin might function directly on BRAF activity or in combination with PP2A to regulate the phosphorylation state of BRAF to control neuronal cell divisions. BRAF, as opposed to AMPK and mTOR, might be the primary target of metformin for controlling neuroblastic cell proliferation. In this regard, the further study of metformin function might help reveal the acute mechanisms involved in this drug’s ability to evoke anticancer signal transduction in animals.

## 2. Results

### 2.1. Metformin Suppresses the Q-Cell Divisions Occurring in AMPK-Deficient Worms during L1 Arrest

The neuronal Q-cell lineage in *C. elegans* only has two mother cells in newly hatched L1 worms: QR and QL—Figure 1A,C. These two cells undergo divisions to generate six final postembryonic neuron cells (A/PVM, A/PQR, and SDQR/L) when worms are fed normally; otherwise, they will remain quiescence when there is no food supply (L1 arrest)—Figure 1A,C. We found that two genes control QR/L proliferation during L1 arrest: *aak-2/ampk* and *daf-18/pten*. Loss of *aak-2* or *daf-18* causes L1 worms to produce a high penetrance of Q-cell divisions [26]. As metformin is considered to regulate AMPK and human insulin resistance, we treated the *aak-2(ok524)* and *daf-18(ok480)* mutants with metformin and found that metformin could significantly suppress the Q-cell divisions in *aak-2(ok524)* L1-arrested mutants (Figure 1D), while metformin had no effect on *daf-18(ok480)* mutants—Figure 1E. As there are two AMPK catalytic subunits (AAK-1 and AAK-2) in *C. elegans*, metformin might work via AAK-1. Next, we assessed *aak-1(tm1944)*; *aak-2(ok524)* double mutants and found that metformin could still suppress the Q-cell divisions even when the functions of *aak-1* and *aak-2* were both abolished—Figure 1F. Additionally, we used an AMPK activator, AICAR, to treat *aak-2(ok524)* and *aak-1(tm1944)*; *aak-2(ok524)* worms and found that AICAR could suppress the Q-cell divisions in *aak-2(ok524)* worms, but it failed to have any effect on the *aak-1(tm1944)*; *aak-2(ok524)* double mutants—Figure 1G. These observations suggested that AICAR functions directly on AMPK. However, we found that metformin could suppress the Q-cell divisions in both the *aak-2(ok524)* and *aak-1(tm1944)*; *aak-2(ok524)* double mutants—Figure 1F,G. We also treated the *aak-2(ok524)* mutants with metformin when the AMPK agonist AICAR was present. We found that metformin can further suppress the Q cell divisions in *aak-2(ok524)* mutants treated with both metformin and AICAR—Figure 1G. Together, these results suggested that metformin might function downstream of AMPK or might be partially dependent on AMPK.

### 2.2. Metformin Suppresses Q-Cell Divisions Independent of mTOR

We found that metformin might function downstream of AMPK. The mTOR pathway has been reported to be a main downstream target inhibited by AMPK [3]. Additionally, metformin has been reported to coordinate the AMPK and mTORC1 pathways in *C. elegans* to control longevity [18]. We assessed whether metformin acts through mTOR to suppress the Q-cell divisions during L1 arrest. First, we tested if metformin affected the phosphorylation level of S6K, a downstream target of mTOR, in an HEK293 cell line. We treated the HEK293T cells with metformin for 24–48 h and the results showed that the S6K phosphorylation level was reduced by metformin treatment—Figure 2A. This result was consistent with previous reports indicating that metformin can regulate S6K function in *C. elegans* [18]. However, we found that mutation of *rsks-1*, the S6K protein-coding gene in *C. elegans*, did not suppress the Q-cell divisions in *aak-2(ok524)* L1-arrested worms—Figure 2F. In contrast, disruption of *rsks-1* can cause the worms to have Q cell divisions (Figure 2B) and increase the percentage of PVM (a Q-cell descendant) appearance—Figure 2C–E.

Next, we tested whether mTOR was involved in regulating Q-cell divisions induced by AMPK deficiency during L1 arrest. We found that RNAi targeting the mTORC1 and mTORC2 genes, *daf-15* and *let-363,* failed to suppress the Q-cell divisions in *aak-2(ok524)* L1 mutants—Figure 2F. Additionally, RNAi targeting of the downstream genes positively regulated by the mTOR pathway, such as *hif-1* [32], *unc-51* [33], and the TORC2 target *sgk-1* [34], also failed to suppress the *aak-2(ok524)* L1-arrest Q-cell division phenotype—Figure 2F. PHA-4/FOXA and SKN-1/NRF are two important transcription factors that are negatively regulated by the TOR pathway [35]; however, we found that *pha-4* and *skn-1* L1-arrested mutants presented with no Q-cell divisions. These results suggested that the TOR pathway was not involved in the *aak-2/ampk* Q-cell division phenotype. Additionally, we treated *aak-1(tm1944)*; *aak-2(ok524)* mutants with the TOR inhibitor rapamycin [36,37] to block mTOR function and we found that rapamycin (100 µM) had no function in regard to Q-cell divisions in *aak-1(tm1944)*; *aak-2(ok524)* L1-arrested mutants—Figure 2G. Furthermore, we treated the *aak-1(tm1944)*; *aak-2(ok524)* worms with metformin (50 mM) and rapamycin (100 µM) at the same time and found that metformin could also suppress the Q-cell divisions (Figure 2G), suggesting that metformin does not act by way of mTOR. Together, these results indicated that metformin does not work through mTOR to suppress the Q-cell divisions occurring during L1 arrest.

As mTOR is reported to regulate germ cell proliferation in L1-arrested *aak-1(tm1944)*; *aak-2(ok524)* worms, we tested the role of metformin on germ-cell proliferation in *aak-1(tm1944)*; *aak-2(ok524)* double mutants. We found that both metformin and rapamycin indeed suppressed germ-cell proliferation during L1 arrest—Figure 3. However, *rsks-1* RNAi could not block the germ cell proliferation in *aak-1(tm1944)*; *aak-2(ok524)* L1-arrested worms—Figure 3. These results demonstrated that metformin blocked germ cell proliferation via mTOR, but *rsks-1* was not involved in this regulation.

### 2.3. Q-Cells Treated with Metformin Arrest at the G1/S Stage In Vivo during L1 Arrest

Metformin can help the cell retain quiescence when AMPK has lost its function. As such, we investigated at which cell stage metformin confers its hold on the cell cycle. Previous reports have shown that metformin likely arrests myeloma cells at the G0/G1 phase by targeting the AMPK and mTOR pathways in vitro [38]. However, our previous work indicated the Q cells are arrested at the G1/S stage during L1 arrest [26]. Moreover, as demonstrated in this current study, the AMPK and mTOR pathways are not directly involved in the action of metformin during starvation. In order to test whether metformin could arrest the cell cycle at the G1/S phase in vivo, we treated *aak-1(tm1944)*; *aak-2(ok524)* L1-arrested worms for 48 h with casticin (10 mM), metformin (50 mM), or hydroxyurea (30 mM), respectively, and then fed the worms for another 24 h. If the Q cells are arrested at the G1/S stage during L1 arrest, the cells in worms treated with the G0/G1 blocker casticin will progress through one additional cell cycle when food is added; the cells in worms treated with the G1/S blocker hydroxyurea will remain arrested and not proceed through any phase of the cell cycle when food is added. We found that the casticin-treated worms presented with four cells (Figure 4C) and the hydroxyurea treated worms presented with two cells—Figure 4A. For the metformin-treated worms, about 40% presented with two cells, 60% with six cells, and 0% with four cells—Figure 4B,D. These results suggested that metformin performs similarly to hydroxyurea, indicating that metformin can arrest the Q cells at the G1/S stage in living L1-arrested worms.

### 2.4. Metformin Regulates BRAF Function during L1 Arrest

This study indicated that metformin does not work through mTOR to suppress neuronal Q-cell proliferation during L1 arrest. More likely, metformin acts downstream of AMPK. Previously, we showed that AMPK acts on PP2A and RAF to control Q-cell divisions during L1 arrest [26]. *pptr-2* is the main PP2A gene to regulate Q-cell divisions [26]. We found that metformin treatment could further suppress the Q-cell divisions in *aak-2(ok524)*; *pptr-1(tm3103)* and *aak-2(ok524)*; *pptr-2(ok1467)* mutants Figure 5A. These results suggest that metformin may work on BRAF to regulate Q-cell divisions. AMPK regulates BRAF to control mouse cell proliferation [29] and PP2A can function downstream of AMPK to activate BRAF to control Q-cell proliferation during L1 arrest [26]. With this in mind, we tested whether mutation of BRAF could block the suppression function of metformin in regard to Q-cell divisions. We found that metformin treatment failed to further suppress the Q-cell divisions in *aak-2(ok524)*; *lin-45(n2018)* mutants—Figure 5B. LIN-45 is an ortholog of human BRAF and BRAF is known to be regulated by AMPK to control cell proliferation [29]. BRAF Ser729 is critical for attenuation of ERK signaling by AMPK and we tested the phosphorylation level of Ser729 in HEK297 cells. The results of this analysis indicated that the phosphorylation level of Ser729 was significantly reduced following metformin treatment—Figure 5C.

### 2.5. Metformin Blocks BRAF-MAPK to Control Neuronal Q-Cell Divisions during L1 Arrest

BRAF-MAPK has been reported to regulate Q-cell and germ-cell proliferation in *C. elegans* [26,28,39], and these results suggest that the BRAF-MAPK pathway might be regulated by metformin to control cell divisions. We assessed whether metformin was dependent on BRAF-MAPK to regulate cell divisions in *C. elegans* during L1 arrest. To this end, we treated *aak-2(ok524); mek-2(n1989)* and *aak-2(ok524); mpk-1(ga119)* mutants with metformin. The results indicated that loss of *mek-2/mek* and *mpk-1/erk* could suppress the Q-cell divisions in *aak-2(ok524)* worms. Importantly, metformin had no additional suppression effect on the Q-cell divisions in the *aak-2(ok524); mek-2(n1989)* and *aak-2(ok524); mpk-1(ga119)* mutants—Figure 6A. These results indicated that metformin acted on BRAF to regulate the downstream signal in the MAPK pathway. To further confirm the conclusion that metformin can regulate ERK/MAPK function, we tested the ERK/MAPK phosphorylation levels in metformin-treated HEK293T cells and found that the phosphorylation level of Thr202/Tyr204 was reduced in these cells—Figure 6B. These results indicated that MPK-1/MAPK was a major target regulated by metformin via the BRAF-MAPK pathway in the control of Q-cell divisions during L1 arrest.

## 3. Discussions

Metformin is used as a medicine to treat type 2 diabetes and is considered to be effective against many cancers, including colon, pancreas, breast, liver, endometrium, rectal, urinary, and bladder cancer [40]. However, additional risk for each cancer in patients with diabetes is variable (colon30%, pancreas 50%, breast 20%) and the mechanisms by which each cancer is influenced by type 2 diabetes are also not exactly the same. As such, it is possible that metformin might control different signaling pathways to influence these cancers. Metformin is widely considered to regulate the LKB-AMPK and mTOR pathways; however, its function in neuronal cancer cell proliferation is beyond AMPK and mTOR [23]. In this work, we used a worm model to show that neuronal cell proliferation can be suppressed by metformin and that this is independent of the traditional AMPK and mTOR pathways—Figure 6C. It was shown that metformin acts downstream of AMPK and uses the BRAF-MAPK pathway to control neuronal cell proliferation during L1 arrest. Our study also demonstrated that metformin could induce other ontogenetic transductions when the animals are in different inner metabolism situations and harsh environments, in this case like starvation. Importantly, these findings might help promote the study of the mechanisms by which other anticancer drugs act under different physiological and/or pathological conditions.

In patients with type 2 diabetes, insulin resistance and hyperinsulinemia are considered risk factors for cancer development [23]. In addition, hyperglycemia-related oxidative stress and accumulation of advanced glycation end products might enhance the risk of malignant transformation [23,41]. The ability of metformin to reduce hepatic gluconeogenesis and increase peripheral glucose utilization is related to the insulin pathway [42]. In our work, we found that metformin was not able to suppress the Q-cell divisions in *daf-18(ok480)* mutants and *daf-18* is a human PTEN homology that has been shown to negatively regulate the insulin pathway in *C. elegans* [43]. The link between metformin and insulin usage in this work is apparently absent. However, many reports have shown that the concentrations of metformin used in the animal might result in different consequences. For example, high levels of metformin in the intestinal wall exert insulin-independent effects, whereas liver and muscle tissues exposed to lower concentrations of metformin alter their post-receptor insulin signaling pathways and redirect energy-generating and storage pathways [1,40]. Additionally, growth factor receptor-bound protein 2 (GRB2) might directly link the insulin receptor to the RAS-RAF-ERK pathway to drive cell growth [1,44]. Therefore, this pathway can bypass the downstream signals in the insulin pathway, such as PTEN and AKT, to regulate cellular metabolism, which is a typical feature of tumor cells. Whether metformin works on insulin ligases, the insulin receptor, or growth factor receptor-bound proteins to coverage on the BRAF-MAPK pathway is a topic to be studied in the future. There are two genes coding AMPK in *C. elegans*: *aak-1* and *aak-2*, but only loss of *aak-2* can cause L1 arrest in worms with neuronal cell divisions [26,45]. However, loss of both *aak-1* and *aak-2* at the same time can enhance the cell divisions phenotype. This suggests that *aak-2* plays the main role in controlling neuronal cell growth in L1 arrest. AAK-1 is likely to play the primary role in maintaining germ cell growth, as *aak-1*, but not *aak-2*, induces germ cell divisions in L1-arrested and adult worms [28,45]. However, the question of what is the true function of *aak-1* on neuronal cell growth is still not well answered. Considering that the metformin can suppress the neuronal cell growth when AMPK is not functional, and that AMKP is reported to have an important role in controlling neuronal metabolism, neuroinflammation, and neurodegeneration [46,47,48], the potential use of metformin in neuronal disease may need to be broadly investigated.

The mTOR pathway is clearly not involved in the neuronal cell divisions detailed in this study. In contrast, the S6K gene has demonstrated a role in the generation of PVM cells. Theoretically, S6K should have no function in Q-cell divisions, as the mTOR genes are not involved in the Q-cell divisions or the effects of metformin. Assuming that the mTOR pathway regulates the Q-cell divisions in AMPK-deficient worms, the loss-of-function of *rsks-1/s6k* should suppress these divisions; however, the results indicated that *rsks-1* mutants increased the penetrance of PVM cells. This contradiction also needs to be addressed in future studies. Meanwhile, there is one report indicating that inhibition of insulin growth factor receptor DAF-2 or RSKS-1 (S6K) has an additional effect on the lifespan extension in *C. elegans* [49], suggesting that the longevity produced by *daf-2* and *rsks-1* is mediated by positive feedback regulation of DAF-16 via AMPK. RSKS-1 might also be involved in microRNA regulation to control AMPK activity [50]. As such, the connection between RSKS-1/S6K and AMPK, the insulin pathway, and cancer-related mircoRNAs is also worth investigating.

Metformin has been reported to regulate the G0/G1 cell-cycle arrest in myeloma cell lines in vitro by inducing autophagy via activated AMPK and repressing both the mTORC1 and mTORC2 signaling pathways via *p*-4EBP1, *p*-AKT, and TSC2 [38]. Importantly, this mechanism is dependent on AMPK. However, in our study, the AMPK and mTOR pathways were apparently not the direct target of metformin in relation to suppressing the Q-cell divisions during L1 arrest. During L1 arrest, the germ cells and Q cells are arrested at G1/S [26,51]. Our work indicated that BRAF-MAPK was the main target of metformin during L1 arrest. The MAPK family is reported to regulate cell-cycle arrest and functions to integrate signals from a wide and diverse range of mechanisms to induce cell proliferation and development. The G1/S phase arrest is regulated by MAPK [52], while the mechanisms by which MAPK acts are not quite the same. MAPK increases cyclinD1 promoter activity and protein levels to keep the cells arrested in the S and G2/M phases [53]. CDK and MAPK modify shared sites and synergistically regulate signaling dynamics via a phosphorylation region on the scaffold protein Ste5. Such collaborative phosphorylation can broaden regulatory inputs and diversify the output dynamics of signaling pathways involved in blocking the cell from phase G1 to S [54]. The RAS-MAPK signaling pathway can also be regulated by microtubule stability and the cyclin-dependent kinase inhibitor p27^kip1^ to control the G1/S-phase transition in a spatiotemporal manner [55]. MAPK and the activator protein-1 (AP-1) signaling cascades can also control the transition of cells from the G1 to S phase and alter the expression level of cell-cycle-related gene transduction and transcription [27]. In our work, we demonstrated that the BRAF-MAPK pathway was regulated by metformin when AMPK activity was absent in L1-arrested worms—Figure 6C; however, the mechanisms by which BRAF/MAPK regulate the cell cycle, including which and how many transcription factors respond to metformin induction and how they are regulated by MAPK to control the cell cycle under starvation, dietary restriction, and normal feeding conditions, still need to be carefully addressed.

## 4. Materials and Methods

### 4.1. Strains

Strains were acquired from the Caenorhabditis Genetics Center (CGC) and crossed into *Pegl-17-mCherry (rdvIs1)* or *Pmec-4: GFP (zdIs5)* to visualize the Q cells and their descendants. Standard culture methods were used as previously described [56]. Strains were maintained on worm NGM plates and cultured with OP50 *Escherichia coli* at 20 °C. The following strains were used in this study: Wildtype: N2, TU3401: *sid-1(pk3321) V; uIs69*, CZ10175: *zdIs5*, RDV55: *rdvIs1*, RB712: *daf-18(ok480)*, RB754: *aak-2(ok524)*, AGD397: *aak-1(tm1944); aak-2(ok524)*, RB1206: *rsks-1(ok1255)*, JH2787: *pptr-1(tm3110)*, RB1338: *pptr-2(ok1467)*, SD420: *mpk-1(ga119)/dpy-17(e164) unc-79(e1068)*, MT8666: *mek-2(n1989),* and MU48: *lin-45(n2018) dpy-20(e1282)*.

### 4.2. Reagents and Treatments

Metformin, rapamycin, casticin, and hydroxyurea were bought from Sigma-Aldrich (Darmstadt, Germany). Metformin and hydroxyurea were dissolved in water, while the others were dissolved in ethanol. Serial dilutions of each chemical were added to M9 right after the embryos were prepared, so the worms were hatched in a drug treatment environment and influenced by the chemical prior to L1 arrest.

### 4.3. Analysis of Q-Cell Divisions

Normally fed mixed-staged worms were harvested and bleached to prepare the embryos, as previously described [26]. In brief, the embryos were maintained and hatched in sterile M9 and incubated at 20 °C with low-speed rocking to initiate L1 arrest. The Q-cell descendants were observed under an Axioplan fluorescent microscope (Zeiss, Germany) after 12 h in L1 arrest. A total of 50–200 μL M9 containing L1-arrested worms were removed from the culture at each time point to make sure the sample was larger than 50 worms. The total number of worms and the worms with Q-cell divisions were counted. Q-cell division was confirmed by lineage analysis using the Q-cell marker *Pegl-17: mCherry* (*rdvIs1*) or Q-cell decedents A/PVM marker *Pmec-4: GFP* (*zdIs5*).

### 4.4. Antibody Staining

Antibody staining was performed as previously described [57]. In brief, L1 worms were collected in 100 µL M9, fixed with 200 µL cold 2× witches brew and 100 µL 10% paraformaldehyde and then incubated at 4 °C for 30 min to overnight. The worms were washed twice in Tris-Triton buffer, incubated in 1% βME/Tris-Triton for 1–2 h at 37 °C, washed in 1× Borate buffer, and incubated in 10 mM DTT/1× Borate buffer for 15 min at room temperature. Subsequently, the worms were washed in 1× Borate buffer, incubated in 0.3% H_2_O_2_/1× Borate buffer for 15 min, incubated for 15 min in PBST-B, and then washed with PBST-A. The worms were detected using an Axioplan fluorescent microscope (Zeiss, Germany).

To detect the germline cells, rabbit anti-PGL-1 (P-granule component) (1:20,000) was used as the primary antibody. Detection was performed using an FITC-labeled goat anti-rabbit secondary antibody (1:100). The total number of worms and the worms with germ-cell divisions were counted. Analysis of the worms was performed using an Axioplan fluorescent microscope (Zeiss, Germany).

### 4.5. Cell Culture and Western Blot Analysis

The genes affected by metformin treatment in *C. elegans* have human homology ones in humans. As HEK293T is a stable tool cell line and is also often used to test the protein levels to study metformin functions [3,18,58], HEK293T cell lines were used for Western blot analysis. The cells were grown in DMEM medium supplemented with 10% fetal bovine serum and 1% penicillin-streptomycin in a 5% CO_2_ atmosphere at 37 °C. The cells were then recovered and seeded in six-well dishes at a density of 5 × 10^5^ cells per well with medium containing 50 mM metformin for another 24 and 48 h. The cell lysates were then prepared by boiling in RIPA Lysis Buffer (Thermo, Waltham, MA, USA) added with Protease and Phosphatase Inhibitor Cocktail (Abcam, Cambridge, UK) for 10 min, followed by incubation on ice. The protein concentration was determined. SDS-PAGE was conducted on the same day that the lysates were made and this was followed by electrophoretic transfer to nitrocellulose membrane at 300 mA for 1.5 h on ice. The Western blot results shown are representatives of at least two independent biological replicates. The Phospho-B-Raf (Ser445), Phospho-p70 S6 Kinase (Thr389), and Phospho-p44/42 MAPK (Erk1/2) (Thr202/Tyr204) were all purchased from Sigma.

### 4.6. RNAi

An RNAi bacterial strain (HT115α) expressing a double-stranded target RNA (vector, L4440) was cultured and used to inactivate the target gene function. The mutant worms used to do the RNAi experiment were crossed into the TU3401 strain, so all these worms have *uIs69*, which makes the feeding RNAi work in neuron systems in these worms [59]. Eggs from RNAi fed worms were transferred to fresh NGM plates containing the same bacteria and allowed to grow at 15 °C for 3 days. Larva stage 4 worms were then transferred to the same RNAi NGM plates and cultured at 20 °C for 24 h. The RNAi NGM plates contained 1 mM isopropyl-B-D-thiogalactopyranoside (IPTG) for the induction of the double-stranded RNA. The expression levels of the target genes are sufficiently knocked down by this RNAi method—Appendix A.

## Figures and Tables

**Figure 1 ijms-22-00178-f001:**
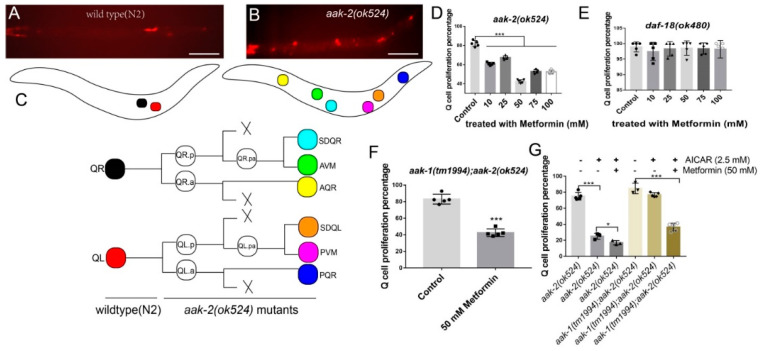
Metformin suppresses neuronal Q-cell divisions in AMP-activated protein kinase (AMPK) deficient L1-arrested worms. (**A**) Wildtype (N2) L1-arrested worms only have two Q cells. (**B**) *aak-2(ok524)* mutants show Q-cell divisions during L1 arrest and all six Q-cell decedents are present. (**C**) Q cells in wildtype (N2) and *aak-2(ok524)* L1-arrested worms; the colors of the cells in the cartoon worms are the same in the cell lineage figure. (**D**) Here, 50 mM metformin has the strongest suppression towards Q-cell divisions in *aak-2(ok524)* mutants during L1 arrest. (**E**) Metformin has no effect on Q-cell divisions in *daf-18(ok480)* mutants during L1 arrest. (**F**) Metformin has a suppressive function on Q-cell divisions, even when AMPK is totally abolished. Metformin can suppress the Q-cell divisions in *aak-1(tm1944)*; *aak-2(ok524)* double mutants. (**G**) Metformin works differently from AICAR, suggesting it may not work directly on AMPK. AICAR is an AMPK activator and works directly on AMPK. AICAR has no effect on *aak-1(tm1944)*; *aak-2(ok524)* double mutants, while metformin can further suppress the Q-cell divisions in rapamycin-treated *aak-1(tm1944)*; *aak-2(ok524)* double mutants. Data are the average of five independent experiments. Error bars: standard deviation (SD). *: *p* < 0.05, ***: *p* < 0.001. Scale bars: 50 μm.

**Figure 2 ijms-22-00178-f002:**
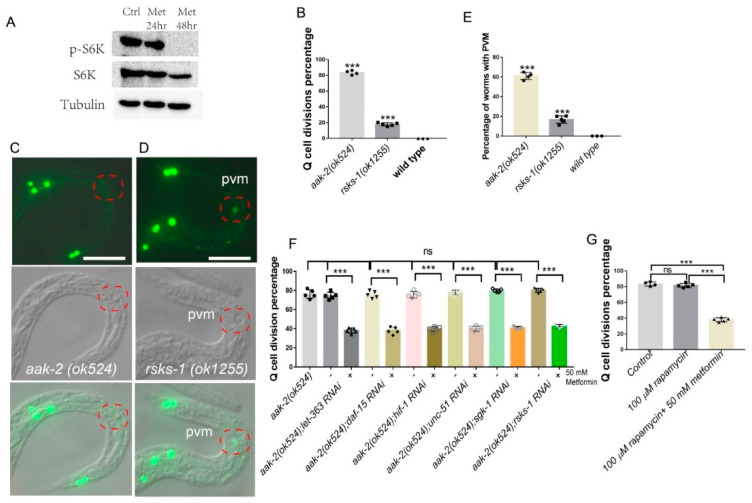
The effect of metformin on Q-cell divisions is independent of mTOR. (**A**) Western blot results depicting S6K phosphorylation levels in metformin-treated H293 cells. Phospho-p70 S6 kinase (Thr389) was significantly reduced following metformin treatment. Ctrl: control, Met: 50 mM metformin. (**B**) *rsks-1* mutants show Q-cell divisions during L1 arrest. (**C–E**) *zdIs5* GFP is a marker for identifying the two decedents of the Q-cell linage; disruption of *rsks-1* causes L1-arrested worms to present with a higher incidence of PVM cells (**E**), a Q-cell decedent, when using *zdIs5* as a marker to study Q-cell divisions. More than 80% of *aak-2(ok524)* worms undergo Q-cell divisions, while only 60% present PVM when using *zdIs5*. All *rsks-1(ok1255)* mutants that show Q-cell divisions have PVM. (**F**) Disruption of mTOR and its target genes cannot suppress the Q-cell divisions during L1 arrest. Metformin can further suppress the Q-cell divisions subsequent to RNAi of these genes. -: no metformin treatment, +: 50 mM metformin was added. RNAi works sufficiently (Appendix A). (**G**) Metformin suppresses the Q-cell divisions in *aak-1(tm1944)*; *aak-2(ok524)* double mutants; rapamycin has no effect on Q-cell divisions in *aak-1(tm1944)*; *aak-2(ok524)* double mutants. Data are the average of 3–5 independent experiments. Error bars: standard deviation (SD). ***: *p* < 0.001. Scale bars: 50 μm.

**Figure 3 ijms-22-00178-f003:**
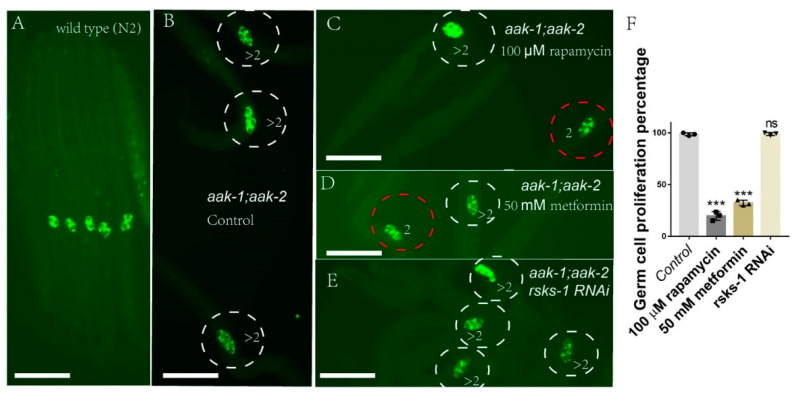
Metformin can suppress germ-cell proliferation during L1 arrest through S6K independent mTOR. Germ-cell-specific antibody (anti-PGL-1) staining. (**A**) Wild-type worms only show two germ cells during L1 arrest. (**B**) *aak-1(tm1944)*; *aak-2(ok524)* double mutants serve as a control and these worms show 100% germ cell proliferation during L1 arrest. *aak-1(tm1944)*; *aak-2(ok524)* double mutants treated with rapamycin (**C**), metformin (**D**), or *rsks-1* RNAi (**E**). Rapamycin and metformin can suppress germ cell proliferation, while *rsks-1* RNAi has no effect on germ-cell proliferation (**F**). Data are the average of three independent experiments. The sample size in each experiment is larger than 80 worms. Error bars: standard deviation (SD). ***: *p* < 0.001. Scale bars: 50 μm.

**Figure 4 ijms-22-00178-f004:**
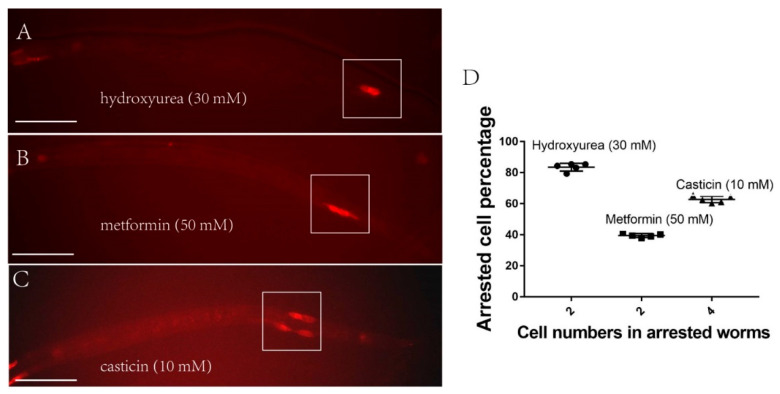
Metformin blocks Q-cell division at the G1/S stage of the cell cycle. Casticin is a G0/G1 blocker and hydroxyurea is a G1/S blocker. (**A**) Hydroxyurea-treated (30 mM) L1-arrested worms after feeding cannot go through the next cell cycle, so only two Q cells are apparent. (**B**) About 40% of metformin-treated (50 mM) L1-arrested worms after feeding cannot go through the next cell cycle, so only two Q cells are apparent. Metformin cannot fully suppress the Q-cell divisions in L1-arrested worms and about 60% of these worms can bypass the function of metformin and go through all the cell divisions (presenting all six Q-cell decedents). (**C**) Most of the casticin-treated (10 mM) L1-arrested worms after feeding can go through one more cell cycle, so four Q cells are apparent. (**D**) Metformin and hydroxyurea hold the Q cells at the G1/S phase and casticin holds the Q cells at the G0/G1 phase. Data are the average of five independent experiments. The sample size in each experiment is larger than 80 worms. Error bars: standard deviation (SD). Scale bars: 50 μm.

**Figure 5 ijms-22-00178-f005:**
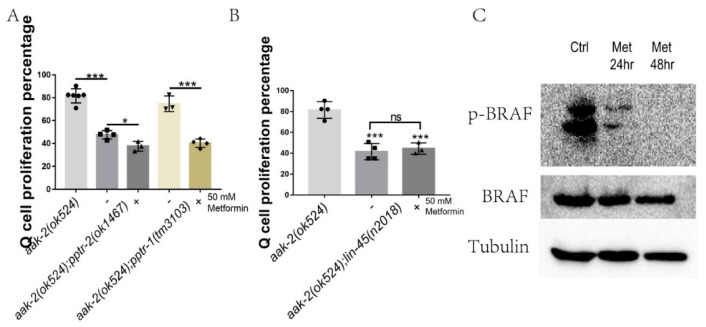
Metformin works on BRAF. (**A**) Metformin suppresses Q-cell divisions in the *pp2a (pptr-1/2)* mutant background. -: no metformin treatment, +: 50 mM metformin was added. (**B**) *lin-45*/*BRAF* mutants block the suppressive effect of metformin on Q-cell divisions in *aak-2(ok524)* mutants. -: no metformin treatment, +: 50 mM metformin was added. Data are the average of at least three independent experiments. Error bars: standard deviation (SD). *: *p* < 0.05, ***: *p* < 0.001. (**C**) Western blot analysis of BRAF phosphorylation level in cells treated with metformin. BRAF phosphorylation activity is reduced by metformin treatment. Ctrl: control, Met: 50 mM metformin.

**Figure 6 ijms-22-00178-f006:**
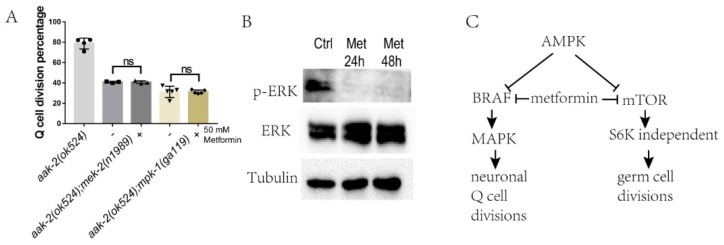
Metformin suppresses Q-cell divisions through BRAF-MAPK. (**A**) *mpk-1/MAPK* mutation blocks the suppression effect of metformin on Q-cell divisions in *aak-2(ok524)* mutants. -: no metformin treatment, +: 50 mM metformin was added. Data are the average of 3–5 independent experiments. Error bars: standard deviation (SD). (**B**) Western blot analysis of ERK/MAPK phosphorylation levels in cells treated with 50 mM metformin. ERK/MAPK phosphorylation activity is reduced by metformin treatment. Ctrl: control, Met: 50 mM metformin. (**C**) A working model depicting the effect of metformin in L1-arrested worms. Metformin works through BRAF to block MAPK function in AMPK-deficient L1-arrested worms to suppress neuronal Q-cell divisions and might also work through mTOR to block germ-cell proliferation; however, the pathways by which metformin regulates cell divisions in these two tissues are distinct.

## Data Availability

The data presented in this study are available in the article or Appendix A.

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
