# Peer review of "BRAF Controls the Effects of Metformin on Neuroblast Cell Divisions in C. elegans"

_ijms, 2020, doi:10.3390/ijms22010178_

Round 1

Reviewer 1 Report

In this work, author present evidence to demonstrate that metformin, thought to mainly act by AMPK activation, has a different target involved in the suppression of Q-cell proliferation in C elegans, providing new insights in the role of BRAF as part of the metformin pathway activation. Experiments are well executed, and results are interesting and relevant, however some improvements are needed.

Major concerns:

It is somewhat difficult to follow the sequence of experiments with the different worm strain, please identify   the ko mutants as such, since it seems that some time aak-2 is used to refer the gene and in other cases as the aak-2 ko mutant. I suggest to use aak-2 ko or -/- to identify null mutants, for instance you said: “Additionally, we used an AMPK activator, AICAR, to treat aak-2 and aak-1; aak-2 worms and found that AICAR 116 could suppress the Q-cell divisions in aak-2 worms, but it failed to have any effect on the aak”…besides, this nomenclature differs from that used in figures legends. This is true for rest of the mutant used in the work.

You are showing SK6 activation in HEK cells, please justify the use of this cells instead of showing Sk6 activation on the worms. Also, AMPK activation by metformin may be assessed under agonists treatment.

Authors conclude that metformin is able to arrest Q-cell division by a pathway independent of AMPK pathway, however, AMPK activation still is responsible for this suppression, however, there is no evidence of the extent of contribution of the AMPK-independent effect versus dependent one, can authors show suppression of Q-cell proliferation under these two conditions? since figure 1 only shows the metformin effect on mutant strains, the effect of metformin on N2 strain is missed. The potential role of aak-1 is also absent in the discussion.

Minor comments

Indicate the strain of C elegans used as wild type.

Line 137:  something is missed in the phrase: “ In contrast, rsks-1 mutants presented with Q-134 cell divisions (Figure 2B), and disruption of rsks-1 demonstrated a high percentage of PVM”

Please provide details of concentrations used for treatment such as rapamycin, casticin, and hydroxyurea.

Include a discussion about the relevance of your findings on clinical fields such as neurodegenerative diseases, to enhance the meaning of the results.

Author Response

point-by-point response to the reviewer's comments:

We thank all reviewers for giving us these wonderful suggestions! We did our best to address all the reviewers’ concerns and suggestions. We sincerely hope that our response and work can satisfy all these concerns pointed out by reviewers.

Reviewer #1

Major concerns:

It is somewhat difficult to follow the sequence of experiments with the different worm strain, please identify   the ko mutants as such, since it seems that some time aak-2 is used to refer the gene and in other cases as the aak-2 ko mutant. I suggest to use aak-2 ko or -/- to identify null mutants, for instance you said: “Additionally, we used an AMPK activator, AICAR, to treat aak-2 and aak-1aak-2 worms and found that AICAR 116 could suppress the Q-cell divisions in aak-2 worms, but it failed to have any effect on the aak”…besides, this nomenclature differs from that used in figures legends. This is true for rest of the mutant used in the work.

Author response: We thank the reviewer for this wonderful suggestion. As not all the mutants used in this work are knockout strains, and also in order to keep consistence in the whole manuscript, figures and figure legends, so we put the details of alleles after the mutant genes. Now the information of all these gene mutations are updated, please check the revised manuscript.

You are showing SK6 activation in HEK cells, please justify the use of this cells instead of showing Sk6 activation on the worms.

Author response: H293T and H293E are very useful tool cell lines, and the protein levels can be properly analyzed in these cell lines, especially the proteins tested in our experiments such as BRAF, S6K, RSKS-1 and other proteins in AMPK and TOR pathways. When H293T or H293E cell lines are treated with metformin, these protein levels can be experimentally measured by using western blots. Considering that the conservational reality of these proteins in human cell lines and C. elegans, the H293T or H293E are always used to test the human homology proteins after the results are found in C. elegans. Metformin functions found in C. elegans are also always tested in H293T or H293E cell lines, such as these works published previously [1-3]. Why human cell lines are usually used to confirm the findings in C. elegans studies? The reason is that antibodies of many proteins in C. elegans are not commercially available currently. Some researchers may use the human antibodies to directly test the homology protein levels in C. elegans, however, the qualities of results yielded by these experiments are always poor. As in our case, we also indeed tested those protein levels in worms by using antibodies descripted in our work, but the protein bands in western blots are difficult to analysis. So the results genetically found in C. elegans are always further confirmed by protein experiments in human cell lines. And the results of protein levels in human cell line experiments firmly consist with the results found in genetic experiments in C. elegans.

We put the sentence “The genes affected by metformin treatment in C. elegans have human homology ones in human. As HEK293T is a stably tool cell line, and also often used to test the protein levels to study metformin functions, so the HEK293T cell lines were used for Western blot analysis.” in the method to justify. Please see the western blots method in the revised manuscript (line341-343 in revised manuscript with tracking changes)! 

Also, AMPK activation by metformin may be assessed under agonists treatment.

Author response: We thank the reviewer for this wonderful suggestion! Metformin likely inhibits the mitochondrial electron transport chain Complex I resulting in changes to the AMP/ATP ratio, which can activate the AMP/ATP energy-sensor AMPK signal pathway [4, 5]. Alternatively, the AMPK pathway might also be activated from upstream by metformin via the LKB1 protein [6, 7]. So for the use of metformin in metabolic disease such as type 2 diabetes, AMPK activation by metformin is widely tested during nearly 60 years studies. However, in our work, we think that the AMPK is not the direct target of metformin to block cell divisions. This is the new important finding of the function of metformin on cell growth and proliferation. We tested the Q cell divisions when AMPK was fully abolished (as in our case, C.elegans AMPK coding genes aak-1 and aak-2 are both mutated and lost function.), we found that the Q cell divisions are blocked by metformin (please see Figure 1F), these results suggest that metformin works on the downstream targets, but not directly work on AMPK in the case of cell growth. We also used the AMPK agonist AICAR to test our finding, our results show that metformin can block Q cell divisions even when AICAR was added (please see Figure 1G), which further confirms our point.

In order to further address the reviewer’s concern, we treat the aak-2 single mutants with metformin when the AMPK agonist AICAR is present. In this case, as aak-1 is still functional in aak-2 signal mutants, so the AMPK agonist AICAR can work on AAK-1 to activate AMPK. We find that metformin can further suppress the Q cell divisions in aak-2 mutants treated with metformin and AICAR, which further confirm that metformin is indeed working on downstream target, not AMPK, to suppress Q cell divisions. We added this new result in Figure 1G, please check (Line121-124 in revised manuscript with tracking changes)!

Authors conclude that metformin is able to arrest Q-cell division by a pathway independent of AMPK pathway, however, AMPK activation still is responsible for this suppression, however, there is no evidence of the extent of contribution of the AMPK-independent effect versus dependent one, can authors show suppression of Q-cell proliferation under these two conditions?

Author response: There are two genes coding AMPK in C. elegans: aak-1 and aak-2. Q cells divide only when aak-2 is lost [8]. AMPK activation is still responsible for the suppression of Q cell divisions; because the AMPK activator AICAR can suppress the Q cell divisions in aak-2 signal mutants (please see our new added data in Figure 1G). In aak-2 signal mutants, AICAR can still activate AAK-1, so the downstream signal transduction can be activate to compensate the result of aak-2 loss of function to suppress the Q cell divisions. However, metformin is working on AMPK independent manner to suppress the Q cell divisions, because that we find metformin can suppress Q cell divisions even when aak-1 and aak-2 are both lost at the same time (please see the data in Figure 1F and G). And loss of BRAF can fully block the function of metformin on Q cell divisions (please see the data in Figure 5B). Metformin suppress Q cell divisions is not directly dependent on AMPK, but dependent on BRAF, which is working downstream of AMPK, is the whole new point of our work.

We did these experiments suggested by reviewer, Please check our results presented in Figure 1F, G and Figure 5B.

since figure 1 only shows the metformin effect on mutant strains, the effect of metformin on N2 strain is missed.

Author response: N2 worms have no Q cell divisions during L1 arrest. Until now, only daf-18 and aak-2 mutants can have Q cell divisions during L1 arrest. So there is no Q cell divisions phenotype can be analyzed in L1 arrest N2 worms. The results present the difference of Q cell divisions between the aak-2 mutants without metformin treatment and the mutants with metformin treatment. And also, N2 worms and N2 worms treated with metformin have no difference in the case of Q cell divisions (0%).  

The potential role of aak-1 is also absent in the discussion.

 Author response: As we discussed above, there are two genes coding AMPK in C. elegans: aak-1 and aak-2. But, only loss of aak-2 can cause L1 arrest worms have cell divisions [8, 9]. However, loss of both aak-1 and aak-2 at the same time can enhance the cell divisions phenotype. This suggests that the aak-2 plays the main role in controlling cell growth in L1 arrest. What is the true function of aak-1 on neuronal cell growth is still not well answered; even many labs (such as Fukuyama Lab, Ryan Baugh lab, Richard Roy lab, Ian Chinsang Lab and ours) have been working on this for many years.

We discussed this in the “Discussion”, please see the revised manuscript (Line252-258 in revised manuscript with tracking changes)!

Minor comments

Indicate the strain of C elegans used as wild type.

Author response: The wild type worms is N2 strain, we indicate this in the main text and figure legends. Please check!

Line 137:  something is missed in the phrase: “ In contrast, rsks-1 mutants presented with Q-134 cell divisions (Figure 2B), and disruption of rsks-1 demonstrated a high percentage of PVM”.

Author response: We carefully read this phrase, now is changed as “In contrast, disruption of rsks-1 can make the worms have Q cell divisions (Figure 2B), and increase the percentage of PVM (a Q-cell descendant) appearance (Figure 2C-E).” Please check the revised manuscript with tracking changes (line 137-139)!

Please provide details of concentrations used for treatment such as rapamycin, casticin, and hydroxyurea.

Author response: We thank the reviewer for this suggestion, now the working concentrations of these drugs are labeled in main text, figures and figure legends.

Include a discussion about the relevance of your findings on clinical fields such as neurodegenerative diseases, to enhance the meaning of the results.

Author response: We thank the review for this wonderful suggestion, now we put some discussion about this in the revised manuscript, please check (line 258-261 in revised manuscript with tracking changes)!

Reference:

  1. Wu L, Zhou B, Oshiro-Rapley N, Li M, Paulo JA, Webster CM, et al. An Ancient, Unified Mechanism for Metformin Growth Inhibition in C. elegans and Cancer. Cell. 2016;167(7):1705-18 e13. Epub 2016/12/17. doi: 10.1016/j.cell.2016.11.055. PubMed PMID: 27984722; PubMed Central PMCID: PMC5390486.
  2. Chen J, Ou YH, Li Y, Hu SM, Shao LW, Liu Y. Metformin extends C. elegans lifespan through lysosomal pathway. eLife. 2017;6. doi: ARTN e31268

10.7554/eLife.31268. PubMed PMID: ISI:000415105400001.

  1. Elgendy M, Ciro M, Hosseini A, Weiszmann J, Mazzarella L, Ferrari E, et al. Combination of Hypoglycemia and Metformin Impairs Tumor Metabolic Plasticity and Growth by Modulating the PP2A-GSK3beta-MCL-1 Axis. Cancer cell. 2019;35(5):798-815 e5. Epub 2019/04/30. doi: 10.1016/j.ccell.2019.03.007. PubMed PMID: 31031016.
  2. Fryer LG, Parbu-Patel A, Carling D. The Anti-diabetic drugs rosiglitazone and metformin stimulate AMP-activated protein kinase through distinct signaling pathways. The Journal of biological chemistry. 2002;277(28):25226-32. Epub 2002/05/08. doi: 10.1074/jbc.M202489200. PubMed PMID: 11994296.
  3. Hawley SA, Gadalla AE, Olsen GS, Hardie DG. The antidiabetic drug metformin activates the AMP-activated protein kinase cascade via an adenine nucleotide-independent mechanism. Diabetes. 2002;51(8):2420-5. Epub 2002/07/30. doi: 10.2337/diabetes.51.8.2420. PubMed PMID: 12145153.
  4. Shaw RJ, Kosmatka M, Bardeesy N, Hurley RL, Witters LA, DePinho RA, et al. The tumor suppressor LKB1 kinase directly activates AMP-activated kinase and regulates apoptosis in response to energy stress. Proceedings of the National Academy of Sciences of the United States of America. 2004;101(10):3329-35. Epub 2004/02/27. doi: 10.1073/pnas.0308061100. PubMed PMID: 14985505; PubMed Central PMCID: PMCPMC373461.
  5. Zhang YL, Guo H, Zhang CS, Lin SY, Yin Z, Peng Y, et al. AMP as a low-energy charge signal autonomously initiates assembly of AXIN-AMPK-LKB1 complex for AMPK activation. Cell metabolism. 2013;18(4):546-55. Epub 2013/10/08. doi: 10.1016/j.cmet.2013.09.005. PubMed PMID: 24093678.
  6. Zheng S, Qu Z, Zanetti M, Lam B, Chin-Sang I. C. elegans PTEN and AMPK block neuroblast divisions by inhibiting a BMP-insulin-PP2A-MAPK pathway. Development. 2018;145(23). Epub 2018/11/30. doi: 10.1242/dev.166876. PubMed PMID: 30487179.
  7. Fukuyama M, Sakuma K, Park R, Kasuga H, Nagaya R, Atsumi Y, et al. C. elegans AMPKs promote survival and arrest germline development during nutrient stress. Biology open. 2012;1(10):929-36. Epub 2012/12/06. doi: 10.1242/bio.2012836. PubMed PMID: 23213370; PubMed Central PMCID: PMC3507181.

Reviewer 2 Report

General comment:

The manuscript “BRAF controls the effects of metformin on neuroblast cell divisions in C. elegans” by Qu et al. reports that C. elegans neuronal cell proliferation can be suppressed by metformin independently of the traditional AMPK and mTOR pathways. This study has presented interesting data linking BRAF to Q cells growth under the metformin actions, which support the conclusion. This information can help promote the understanding of the mechanisms by which other anticancer drugs act under different pathophysiological conditions. The data and writing are excellent; however, some issues should be addressed before presenting a potentially acceptable version of the manuscript for reconsideration.

Specific comments

Major comments:

  1. The authors used RNAi feeding in combination of deletion mutant to generate desirable double mutants in C. elegans. However, feeding RNAi is effective in virtually all cells except neuronal cells (Ref: PMID11223248). Neuronal RNAi knockdown occurs only when dsRNAi is produced within neurons. The lack of neuronal response to RNAi can be a potential concern if the readout is based on Q cells proliferation. Hence, the knockdown efficiency of these targets in neurons should be provided, either by qPCR or Western blot analysis, to strengthen the argument that the reduced expression/function of the target is due to the specific and functional RNAi.
  2. Some data labeling presented in this work seem to contain too many text and thus difficult to read, for example, figure 2F, 5B and 6A. The presentation of different conditions (gene knockdown, knockout, drug administration) should be improved.

Minor comments:

The authors should pay careful attention to the editing of the manuscript because there are some typos and inconsistent errors in the manuscript.  For example, in introduction: paragraph 2, line 71: “C. elegans is wildly used as a model…” should be corrected as “C. elegans is widely used as a model……”; similarly in results, line 214 should be corrected as: “Metformin is widely considered to regulate……”. In result section “Metformin regulates BRAF function during L1 arrest”, line 183, 189 and 193, figure 5A, 5B and 5C are misplaced. Please check figure 5 again and correct them. In figure legends, p18, line 524 “three independent experiments” and line 544 “at least 3 independent experiments”, consistency is important throughout the manuscript.

Author Response

point-by-point response to the reviewer's comments:

We thank all reviewers for giving us these wonderful suggestions! We did our best to address all the reviewers’ concerns and suggestions. We sincerely hope that our response and work can satisfy all these concerns pointed out by reviewers.

Reviewer #2

Major comments:

  1. The authors used RNAi feeding in combination of deletion mutant to generate desirable double mutants in C. elegans. However, feeding RNAi is effective in virtually all cells except neuronal cells (Ref: PMID11223248). Neuronal RNAi knockdown occurs only when dsRNAi is produced within neurons. The lack of neuronal response to RNAi can be a potential concern if the readout is based on Q cells proliferation. Hence, the knockdown efficiency of these targets in neurons should be provided, either by qPCR or Western blot analysis, to strengthen the argument that the reduced expression/function of the target is due to the specific and functional RNAi.

Author response: We thank the reviewer to point out this important concern. As you know, feeding RNAi can not work in neurons in C. elegans, because there is no the transmembrane protein SID-1 in the neuron system in C. elegans. SID-1 is essential for systemic RNAi in C. elegans because it allows the passive cellular uptake of dsRNA. We realized this problem at the beginning, so the hypersensitive neuronal feeding RNAi strain :TU3401 sid-1(pk3321) V; uIs69 V was crossed into aak-2 mutants[1], so the aak-2 worms used to do the RNAi experiments are actually containing the transgene uls69 [pCFJ90 (myo-2p::mCherry) + unc-119p::sid-1], which means that the RNAi can work fine in the neuronal system. We mention this in method now, please check the RNAi method part (please see the line357-359 in revised manuscript with tracking changes!). In order to further address this concern, we did the experiments suggested by reviewer, we used qPCR to test the expression levels of target genes. The results show that the target genes are efficiently knockdown. Please see line363-364 and the results in the “Supplementary materials”.

  1. Some data labeling presented in this work seem to contain too many text and thus difficult to read, for example, figure 2F, 5B and 6A. The presentation of different conditions (gene knockdown, knockout, drug administration) should be improved.

 Author response: We agree that the labeling is too complicate to read. Because the double gene mutants were used in this work, so the gene names and allele names together are too long. We accept this suggestion, and we did our best to change the labeling, please check all these figures. We hope the changes can present the results better.

Minor comments:

The authors should pay careful attention to the editing of the manuscript because there are some typos and inconsistent errors in the manuscript.  For example, in introduction: paragraph 2, line 71: “C. elegans is wildly used as a model…” should be corrected as “C. elegans is widely used as a model……”; similarly in results, line 214 should be corrected as: “Metformin is widely considered to regulate……”. In result section “Metformin regulates BRAF function during L1 arrest”, line 183, 189 and 193, figure 5A, 5B and 5C are misplaced. Please check figure 5 again and correct them. In figure legends, p18, line 524 “three independent experiments” and line 544 “at least 3 independent experiments”, consistency is important throughout the manuscript.

Author response: We thank the reviewer for pointing out these typos! Now we corrected them in the revised manuscript!

Again, we thank all the reviewers for giving us so many wonderful suggestions, and we sincerely hope our response can satisfy your concerns. We also welcome any further feedback from you! Thanks again!

Reference:

  1. Calixto A, Chelur D, Topalidou I, Chen XY, Chalfie M. Enhanced neuronal RNAi in C. elegans using SID-1. Nature methods. 2010;7(7):554-U102. doi: 10.1038/Nmeth.1463. PubMed PMID: ISI:000279343300024.

Round 2

Reviewer 1 Report

The authors resolved all the concerns correctly, and the manuscript is now acceptable for publications.

Author Response

Thanks again!

Reviewer 2 Report

The authors have corrected typos and errors in the revised manuscript. Importantly, they have provided a supplementary data which clearly shows the knockdown efficiency by qPCR in the hypersensitive neuronal RNAi strain. However, data labeling of figure 2F, 5A, 5B and 6A still need modification. It is suggested that the rectangular shape outlines in these figures are redundant and should be removed. The strain name can then be moved to the center, beneath the “-” and “+” labels. Lastly, the label of “-” and “+” in this figures can use a drug name labeled at the side.

Author Response

Thanks for giving us these wonderful suggestions! We accepted these suggestions and changed the labeling in Figure 2F, 5A, 5B and 6A. Please see Figure 2, 5 and 6 in the revised manuscript!